# Differentially Private Clipped-SGD: High-Probability Convergence with Arbitrary Clipping Level

## Abstract

Gradient clipping is a fundamental tool in Deep Learning, improving the high-probability convergence of stochastic first-order methods like SGD, AdaGrad, and Adam under heavy-tailed noise, which is common in training large language models. It is also a crucial component of Differential Privacy (DP) mechanisms. However, existing high-probability convergence analyses typically require the clipping threshold to increase with the number of optimization steps, which is incompatible with standard DP mechanisms like the Gaussian mechanism. In this work, we close this gap by providing the first high-probability convergence analysis for DP-Clipped-SGD with a fixed clipping level, applicable to both convex and non-convex smooth optimization under heavy-tailed noise, characterized by a bounded central $\alpha$-th moment assumption, $\alpha \in (1, 2]$. Our results show that, with a fixed clipping level, the method converges to *a neighborhood* of the optimal solution with a *faster rate* than the existing ones. The neighborhood can be balanced against the noise introduced by DP, providing a refined trade-off between convergence speed and privacy guarantees.

## 1 Introduction

Stochastic first-order optimization methods, such as Stochastic Gradient Descent (SGD) (Robbins and Monro, 1951), AdaGrad (Streeter and McMahan, 2010; Duchi et al., 2011), and Adam (Kingma and Ba, 2014), are fundamental for training modern Machine Learning (ML) and Deep Learning (DL) models. However, these methods are often enhanced with additional algorithmic techniques that play a critical role in their convergence and practical performance. Among these, gradient clipping (Pascanu et al., 2013) is one of the most widely used and well-studied approaches. In recent years, substantial efforts have been made to theoretically understand the advantages of gradient clipping and its impact on the convergence of stochastic optimization algorithms.

In particular, gradient clipping is a key component in managing heavy-tailed noise, which commonly arises in the training of language models on textual data (Zhang et al., 2020), in the training of GANs (Goodfellow et al., 2014; Gorbunov et al., 2022), and even in simpler tasks such as image classification (Şimşekli et al., 2019). This approach is primarily analyzed through the lens of high-probability convergence, as such guarantees provide a more accurate reflection of the actual behavior of optimization methods compared to their more conventional in-expectation counterparts (Gorbunov et al., 2020). Moreover, as demonstrated by Sadiev et al. (2023) for SGD and by Chezhegov et al. (2024) for AdaGrad and Adam, methods without clipping may fail to exhibit high-probability convergence with logarithmic dependence on the failure probability. In contrast, several recent works (Gorbunov et al., 2020; Cutkosky and Mehta, 2021; Sadiev et al., 2023; Nguyen et al., 2023; Gorbunov et al., 2024b; Chezhegov et al., 2024; Parletta et al., 2024) have established that various stochastic

first-order methods attain significantly better high-probability convergence under heavy-tailed noise assumptions across different settings.

On the other hand, clipping is a cornerstone of Differentially Private (DP) machine learning. The widely used Gaussian mechanism (Dwork et al., 2014) achieves privacy by adding Gaussian noise to the gradients, thereby introducing uncertainty about their true values. However, the DP guarantees provided by this mechanism rely on the assumption that the gradients have bounded norms, a condition typically enforced through gradient clipping (Abadi et al., 2016).

It is therefore tempting to claim that gradient clipping can provably address two distinct challenges simultaneously: mitigating heavy-tailed noise and ensuring differential privacy (DP). However, this is not entirely accurate, as the clipping policies required for these two objectives differ substantially. In the context of heavy-tailed noise, existing convergence guarantees are typically derived assuming that the clipping level increases with the total number of training steps. In contrast, DP mechanisms require a fixed and bounded clipping threshold to ensure robust privacy guarantees. This fundamental mismatch raises a critical question:

*How does differentially private version of* Clipped-SGD *converge with high probability under the heavy-tailed noise?*

**Our contribution.** In this paper, we address the above question by providing the first high-probability convergence bounds for the differentially private version of Clipped-SGD (DP-Clipped-SGD) with an *arbitrary fixed clipping level* applied to convex smooth optimization problems under heavy-tailed noise. Specifically, we assume that the stochastic gradient has a bounded central $\alpha$-th moment for some $\alpha \in (1, 2]$ and establish that DP-Clipped-SGD achieves a high-probability convergence rate of $\widetilde{\mathcal{O}}(K^{-1/2})$ to a certain *neighborhood* of the optimal solution. This rate is significantly better than the previously known bound of $\widetilde{\mathcal{O}}(K^{-(\alpha-1)/\alpha})$ in this setting.

However, this improvement is achieved by relaxing the requirement for exact convergence and instead demonstrating convergence to a neighborhood whose size depends non-trivially on the clipping level, noise scale, and other problem-dependent parameters. Importantly, the size of this neighborhood, introduced due to the inherent bias in clipped stochastic gradients, can be carefully balanced with the neighborhood induced by the DP noise, allowing for more flexible control over the trade-off between convergence accuracy and privacy. Additionally, we extend our results to the non-convex case, illustrating the broader applicability of our analysis.

## 2 Technical Preliminaries

The optimization problem considered in this work has the following form

$$\min_{x \in \mathbb{R}^d} \left\{ f(x) := \mathbb{E}_{\xi \sim \mathcal{D}}[f_\xi(x)] \right\}. \tag{1}$$

Here, $x$ denotes the model parameters, $f : \mathbb{R}^d \to \mathbb{R}$ is the expected loss function, and $f_\xi : \mathbb{R}^d \to \mathbb{R}$ represents the loss computed for a random sample $\xi$ drawn from an (often unknown) distribution $\mathcal{D}$. Such problems are fundamental in machine learning (Shalev-Shwartz and Ben-David, 2014).

We assume that at each iteration, we have access to an oracle that provides a stochastic gradient $\nabla f_\xi(x)$, as well as a $d$-dimensional random vector $\omega$ sampled from a Gaussian distribution $\mathcal{N}(0, \sigma_\omega^2 \mathbf{I}_d)$, where $\mathbf{I}_d$ is the $d \times d$ identity matrix. More precisely, the random variables $\xi$ and $\omega$ are defined on the probability space $\left( \Omega_d \times \mathbb{R}^d, \mathcal{B}(\Omega_d) \otimes \mathcal{B}(\mathbb{R}^d), \mathcal{F}^t, \mathbb{P} \right)$, where $\Omega_d$ represents the data sample space, and $\mathcal{B}(\mathcal{X})$ denotes the Borel $\sigma$-algebra generated by the set $\mathcal{X}$. This probability space is also equipped with the natural filtration $\mathcal{F}^t = \sigma \left( \left[\nabla f_{\xi^0}(x^0), \omega_0\right]^T, \dots \left[\nabla f_{\xi^t}(x^t), \omega_t\right]^T \right)$, which captures the history of the stochastic process up to time $t$. The probability measure $\mathbb{P}$ is defined as the product measure on this space, given by

$$\mathbb{P}\{B_d \times B_\omega\} = (\mu \times \nu)(B_d \times B_\omega) = \mu(B_d) \, \nu(B_\omega), \quad \forall B_d \in \mathcal{B}(\Omega_d), \forall B_\omega \in \mathcal{B}(\mathbb{R}^d), \tag{2}$$

where $\mu$ is a probability measure on $\Omega_d$, and $\nu$ is the Gaussian measure on $\mathbb{R}^d$ with mean zero and covariance matrix $\sigma_\omega^2 \mathbf{I}_d$.

**Types of convergence bounds.** Several types of convergence bounds are commonly used to analyze the behavior of stochastic optimization methods, ranging from in-expectation bounds to almost sure convergence guarantees. High-probability convergence bounds provide guarantees of the form $\mathbb{P}\left\{\mathcal{P}(x^K) \leq \epsilon\right\} \geq 1 - \beta$, where $\mathcal{P}(x)$ is a performance metric that measures the quality of the solution[1]. Here, $\mathbb{P}\{\cdot\}$ denotes the probability measure defined by the problem setup, $x^K$ is the algorithm's output after $K$ iterations, $\beta$ is the confidence level (or failure probability), and $\epsilon$ is the optimization error.

This type of convergence is generally considered superior to in-expectation guarantees (e.g., $\mathbb{E}[\mathcal{P}(x^K)] \leq \epsilon$), as it captures not only the average behavior of the underlying random variables but also their tail behavior, which is particularly important for distributions with heavy tails. However, it is worth noting that the number of iterations $K$ required to achieve such high-probability guarantees can depend inversely on the failure probability $\beta$, as seen in analyses for methods like SGD (Sadiev et al., 2023), AdaGrad, and Adam (Chezhegov et al., 2024). Such inverse-power dependencies on $\beta$ are generally undesirable, as $\beta$ is typically chosen to be very small. Consequently, a major objective in the high-probability convergence literature is to establish bounds with polylogarithmic dependence on $1/\beta$, which are significantly tighter and more practical.

**Assumptions.** In the following, we list the assumptions on the structure of the problem at hand. These assumptions are very mild and cover a wide range of problems.

**Assumption 2.1.** We assume the function $f$ is uniformly lower-bounded on some subset $Q \subseteq \mathbb{R}^d$, i.e., $f_* := \inf_{x \in Q} f(x) > -\infty$.

The above assumption is necessary for problem (1) to be feasible. Next, we make a standard assumption about the smoothness of the objective function.

**Assumption 2.2.** We assume that there exists a constant $L > 0$ such that for all $x, y \in Q \subseteq \mathbb{R}^d$ the function $f$ satisfies the following.

$$\|\nabla f(x) - \nabla f(y)\| \leq L \|x - y\|. \tag{3}$$

In this work, we consider both classes of convex and non-convex functions. The following assumption holds only for convex functions.

**Assumption 2.3.** We assume there exists a subset $Q$ of $\mathbb{R}^d$ such that for all $x, y \in Q$

$$f(y) \geq f(x) + \langle \nabla f(x), y - x \rangle. \tag{4}$$

The following assumption is with respect to the stochastic oracle that our algorithm receives at each iteration. We assume that the stochastic gradients have a bounded central $\alpha$ moment for some $\alpha \in (1, 2]$. This assumption is stated explicitly below.

**Assumption 2.4.** We assume there exist some subset $Q \subseteq \mathbb{R}^d$, and some constants $\sigma > 0$, $\alpha \in (1, 2]$ such that for all $x \in Q$

$$\mathbb{E}_{\xi \sim D}\left[\nabla f_\xi(x) \mid x\right] = \nabla f(x), \tag{5}$$

$$\mathbb{E}_{\xi \sim D}\left[\|\nabla f_\xi(x) - \nabla f(x)\|^\alpha \mid x\right] \leq \sigma^\alpha. \tag{6}$$

As it can be seen, in the case $\alpha = 2$, the aforementioned conditions recover the standard uniformly bounded variance assumption widely used for obtaining convergence guarantees for optimization algorithms in the literature. Since the $L^p$ norms of random variable are non-decreasing in $p$, this assumption allows the stochastic gradients to have infinite variance.

Next, we use the classical definition of $(\varepsilon, \delta)$-differential privacy. Intuitively, it provides probabilistic guarantees that an intruder cannot infer the existence of a particular data in the data set that the algorithm used to train the model.

**Definition 2.5.** ( $(\epsilon, \delta)$-Differential Privacy (Dwork et al., 2014)). A randomized method $\mathcal{M} : \mathcal{D} \to \mathcal{R}$ satisfies $(\varepsilon, \delta)$-Differential Privacy, if for any adjacent $D, D' \in \mathcal{D}$ and for any $\mathcal{S} \subseteq \mathcal{R}$

$$\mathbb{P}\left(\mathcal{M}(\mathcal{D}) \in \mathcal{S}\right) \leq e^\varepsilon \mathbb{P}\left(\mathcal{M}(\mathcal{D}') \in \mathcal{S}\right) + \delta, \tag{7}$$

Smaller $(\varepsilon, \delta)$ provides stronger privacy guarantee. This also can be viewed from the perspective of Bayesian hypothesis testing where the null and alternative hypothesis are about the existence of an individual's data in the dataset (Su, 2024).

---

[1]Examples of such performance metric for problem (1): $\mathcal{P}(x) = f(x) - f(x^*)$, $\mathcal{P}(x) = \|\nabla f(x)\|^2$, $\mathcal{P}(x) = \|x - x^*\|^2$, where $x^* \in \arg\min_{x \in \mathbb{R}^d} f(x)$.

## 3   Related Work

**Clipping in Differential Private learning.**   There are several approaches to ensuring DP guarantees in SGD, but the most common method relies on a combination of gradient clipping and noise injection. In the finite-sum setting, Abadi et al. (2016) demonstrated that it is sufficient to add Gaussian noise (the Gaussian mechanism) with standard deviation $\sigma_\omega = \Theta\left(\frac{q\lambda}{\varepsilon}\sqrt{K \ln\frac{1}{\delta}}\right)$ to the clipped gradients, where $q$ is the sampling probability for each individual summand. This approach reduces the variance of the required Gaussian noise by a factor of $\sqrt{\ln K}$ compared to the advanced composition theorem (Dwork et al., 2014), significantly improving the utility of DP training.

This combination of gradient clipping and the Gaussian mechanism has become a standard approach in many DP training algorithms. However, these methods often rely on restrictive assumptions, such as requiring the clipping level to always be larger than the norm of the transmitted vector (Zhang et al., 2022; Noble et al., 2022; Allouah et al., 2023, 2024; Li and Chi, 2025)[2], assuming symmetry of the noise distribution (Liu et al., 2022), or requiring that the full gradients be computed (Wei et al., 2020). These conditions can be quite restrictive, particularly in practical large-scale settings.

To the best of our knowledge, the only work that avoids these assumptions is Islamov et al. (2025), where the authors proposed a distributed optimization method based on clipping, error feedback (Seide et al., 2014; Richtárik et al., 2021), and heavy-ball momentum (Polyak, 1964). However, their high-probability convergence analysis critically relies on the assumption that the noise in the stochastic gradients has sub-Gaussian tails. By contrast, under the more realistic Assumption 2.4 with $\alpha \geq 2$ (which is still more restrictive than the heavy-tailed case with $\alpha < 2$), Zhao et al. (2025) derive in-expectation convergence bounds for a variant of projected SGD that uses DP mean estimation with a sufficiently large number of samples. However, this approach can be prohibitively expensive in practice, particularly in the training of large language models.

**High-probability convergence bounds.**   If the noise in the stochastic gradient has light tails, then classical stochastic first-order methods like SGD and its adaptive and momentum-based variants can achieve desirable high-probability convergence rates, characterized by polylogarithmic dependence on the failure probability $\beta$. For instance, under the sub-Gaussian noise assumption, such results exist for SGD (Nemirovski et al., 2009; Harvey et al., 2019), its accelerated variants (Ghadimi and Lan, 2012; Dvurechensky and Gasnikov, 2016), and its momentum and AdaGrad versions (Li and Orabona, 2020; Liu et al., 2023). Additionally, Madden et al. (2024) demonstrate that polylogarithmic high-probability bounds can also be achieved for SGD under the weaker sub-Weibull noise assumption. However, as highlighted by Sadiev et al. (2023) and Chezhegov et al. (2024), methods like SGD, AdaGrad, and Adam can fail to achieve these desired high-probability rates under heavier-tailed noise distributions.

To address the limitations of high-probability convergence for stochastic methods under heavy-tailed noise, several algorithmic modifications have been proposed and rigorously analyzed in recent years. Nazin et al. (2019) introduced a variant of Stochastic Mirror Descent (Nemirovskij and Yudin, 1983) with *truncation* of the stochastic gradient, establishing high-probability complexity bounds for convex and strongly convex smooth optimization over compact sets under the bounded variance assumption (Assumption 2.4 with $\alpha = 2$). Interestingly, the truncation operator used in this work, while not identical, is closely related to the standard *gradient clipping* technique that has since become the foundation of many subsequent studies.

In particular, Gorbunov et al. (2020) derived the first high-probability complexity bounds for Clipped-SGD and also proposed an accelerated version based on the Stochastic Similar Triangles Method (SSTM) (Gasnikov and Nesterov, 2016). These results were later extended to non-smooth problems by Gorbunov et al. (2024a); Parletta et al. (2024), to unconstrained variational inequalities by Gorbunov et al. (2022), and to settings with noise having a bounded $\alpha$-th moment by Cutkosky and Mehta (2021) (with an additional bounded gradient assumption in the non-convex case). Building on these foundations, Sadiev et al. (2023) extended the results from Gorbunov et al. (2020) and Gorbunov et al. (2022) to the more challenging setting defined by Assumption 2.4 with $\alpha < 2$, removing the bounded gradient assumption for non-convex objectives. This work also introduced

---

[2]Li and Chi (2025) also provide an in-expectation convergence result without the bounded gradient assumption, but with a worse dependence on the variance bound of the stochastic gradients.

new high-probability bounds for Clipped-SGD in the non-convex regime. These non-convex results were further refined by Nguyen et al. (2023), who also obtained tighter logarithmic factors in the convergence rates for both convex and strongly convex settings.

In the context of distributed optimization, Gorbunov et al. (2024b) extended the results of Sadiev et al. (2023) to distributed composite minimization and variational inequalities using the clipping of gradient differences, thereby broadening the applicability to decentralized and federated learning scenarios.

Adaptive methods have also been analyzed through the lens of high-probability convergence. Li and Liu (2023) derived new high-probability bounds for Clipped-AdaGrad with scalar stepsizes, while Chezhegov et al. (2024) obtained analogous bounds for various versions of Clipped-AdaGrad and Clipped-Adam with both scalar and coordinate-wise stepsizes. Additionally, Kornilov et al. (2023) proposed a zeroth-order variant of Clipped-SSTM and analyzed it under Assumption 2.4, extending the clipping framework to derivative-free settings.

However, a critical limitation shared by all of these methods is that the clipping level $\lambda$ is typically chosen as an increasing function of the total number of steps $K$[3]. This choice, while theoretically convenient, leads to prohibitively large DP noise variance when aiming to guarantee $(\varepsilon, \delta)$-DP, resulting in utility bounds that grow with $K$ and significantly degrade the practical effectiveness of these methods in privacy-preserving applications.

There exist other alternatives to gradient clipping that also ensure high-probability convergence with polylogarithmic dependency on the failure probability. They include robust distance estimation coupled with inexact proximal point steps (Davis et al., 2021), gradient normalization (Cutkosky and Mehta, 2021; Hübler et al., 2024), and sign-based methods (Kornilov et al., 2025). Notably, the approaches from Hübler et al. (2024); Kornilov et al. (2025) enjoy provable (yet sub-optimal) high-probability convergence even when $\alpha$ is unknown. In the special case of symmetric distributions, Armacki et al. (2023, 2024) provide new high-probability convergence bounds for a large class of SGD-type methods with non-linear transformations such as standard clipping, coordinate-wise clipping, normalization, and sign-operator, and Puchkin et al. (2024) derive high-probability convergence of SGD with median-based clipping and also extend this result to problems with structured non-symmetry for SGD with smoothed median of means coupled with gradient clipping.

# 4 Main Results

The well-known Clipped-SGD algorithm with the Gaussian DP mechanism (DP-Clipped-SGD) is described in Algorithm 1. If differential privacy (DP) is not required, one can simply set $\sigma_\omega^2 = 0$. As shown by Sadiev et al. (2023), achieving exact convergence to the optimal solution of problem (1) using Clipped-SGD requires the clipping level to be chosen as $\lambda = \mathcal{O}\left(\sigma \left(K/(\ln \frac{K}{\beta})\right)^{1/\alpha}\right)$. However, this choice of clipping level, which scales with the total number of iterations $K$, is problematic from a DP perspective. Specifically, larger clipping levels necessitate larger DP noise to maintain privacy, significantly increasing the variance in gradient estimates and leading to a larger convergence neighborhood.

To address this limitation, in this work, we focus on the more general case of arbitrary fixed clipping levels that do not scale with the total number of iterations. This approach is more compatible with practical DP requirements, where clipping levels are typically kept constant. However, our theoretical results can also accommodate clipping levels that scale with $K$ up to this order, as we discuss in detail in the appendix. This broader analysis introduces a few additional step-size conditions, which we also explore thoroughly in the supplementary material.

The following two theorems present our newly derived step-size bounds and the corresponding performance guarantees for both convex and non-convex settings. Following each theorem, we provide a table that further simplify the performance bounds under the assumption that the clipping level falls within specific intervals. In these tables, we assume that no DP noise is present, focusing purely on the impact of the clipping bias. The final corollary extend these results to the case where

---

[3]In some cases, such as the analysis of Clipped-SSTM (Gorbunov et al., 2020) or Clipped-SGD under strong convexity (Sadiev et al., 2023), the clipping level decreases as a function of the current iteration counter $k$ but still increases overall as a function of $K$.

**Algorithm 1** DP-Clipped-SGD

**Input:** starting point $x^0$, number of iterations $K$, stepsize $\gamma > 0$, clipping level $\lambda$.
1: **for** $k = 0, \ldots, K$ **do**
2:    Compute $\hat{g}_k = \texttt{clip}\left(\nabla f_{\xi^k}(x^k), \lambda\right)$ using a fresh sample $\xi^k \sim \mathcal{D}$
3:    $\omega_k \sim \mathcal{N}(0, \sigma_\omega^2 I_d)$
4:    $\widetilde{g}_k = \hat{g}_k + \omega_k$
5:    $x^{k+1} = x^k - \gamma \widetilde{g}_k$
6: **end for**

DP noise is included in the convex case, while the result for DP case in the non-convex setup is deffered to the supplementary materials due to space limitation.

**Convex problems.** We start with the convex case.

**Theorem 4.1** (Convergence of DP-Clipped-SGD for the convex objectives)**.** *Let the integer $K \geq 0$ and $\beta \in (0,1]$ be given. Furthermore, let Assumptions 2.1, 2.2, 2.3, 2.4, hold for $Q = B_{2R}(x^\star)$, $R \geq \left\|x^0 - x^\star\right\|$. Set $\zeta_\lambda := \max\left\{0, 2LR - \frac{\lambda}{2}\right\}$, and further assume that the step-size $\gamma$ is selected to satisfy*

$$
\gamma \leq \mathcal{O}\left( \min\left\{ \frac{1}{L}, \frac{R}{\lambda^{1-\alpha/2}\sqrt{K \ln\left(\frac{K}{\beta}\right)(\sigma^\alpha + \zeta_\lambda^\alpha)}}, \right.\right.
$$

$$
\left.\left. \frac{R\lambda^{\alpha-1}}{K(\sigma^\alpha + \zeta_\lambda^\alpha)\left(\frac{LR}{\lambda} + \frac{\lambda^{\alpha-1}\zeta_\lambda}{\sigma^\alpha + \zeta_\lambda^\alpha} + (\sigma^\alpha + \zeta_\lambda^\alpha)^{\frac{-1}{\alpha}}\right)}, \frac{R}{\sigma_\omega\sqrt{dK \ln\left(\frac{K}{\beta}\right)}} \right\} \right). \tag{8}
$$

*Then, after $K$ iterations of DP-Clipped-SGD, the iterates with probability at least $1 - \beta$ satisfy*

$$
\min_{t \in [0,K]} f(x^t) - f(x^\star) \leq \frac{4R^2}{\gamma(K+1)} + \frac{64LR^4}{\lambda^2\gamma^2(K+1)^2}. \tag{9}
$$

The convergence rate and the neighborhood to which the algorithm converges depend on the magnitude of $\lambda$ in a non-trivial way. Table 1 summarizes these relationships for different values of $\lambda$ in the absence of DP noise. In the special case where $\lambda = \mathcal{O}\left(\sigma\left(K/\ln\frac{K}{\beta}\right)^{1/\alpha}\right)$, our theorem provides a convergence rate of $\mathcal{O}\left(\left((\ln\frac{K}{\beta})/K\right)^{(\alpha-1)/\alpha} + (\ln\frac{K}{\beta})/K\right)$ to the exact solution in the asymptotic regime. This matches the rate previously derived by Sadiev et al. (2023).

In contrast, if $\lambda$ is chosen as a constant, independent of $K$, the leading term in the convergence rate simplifies to $\mathcal{O}(\sqrt{(\ln\frac{K}{\beta})/K})$, which is faster than the more conservative bound $\mathcal{O}\left(\left((\ln\frac{K}{\beta})/K\right)^{(\alpha-1)/\alpha}\right)$. However, this faster rate comes at the cost of only guaranteeing convergence to a neighborhood around the optimal solution, determined by the third term in the stepsize condition (8).

To ensure $(\varepsilon, \delta)$-DP for DP-Clipped-SGD in our setting (i.e., expectation minimization), one can set the noise scale as $\sigma_\omega = \Theta\left(\frac{\lambda}{\varepsilon}\sqrt{K \ln\left(\frac{K}{\delta}\right) \ln\left(\frac{1}{\delta}\right)}\right)$ and apply the advanced composition theorem (Dwork et al., 2014, Theorem 3.22). Given the fourth term in (8), this choice implies that the stepsize decreases as $1/K$, resulting in convergence to a certain neighborhood. This observation is formalized in the next corollary.

**Corollary 4.2** (Convergence of Clipped-SGD for the convex objective)**.** *Let the assumptions of Theorem 4.1 hold, $\sigma_\omega = \Theta\left(\frac{\lambda}{\varepsilon}\sqrt{K \ln\left(\frac{K}{\delta}\right) \ln\left(\frac{1}{\delta}\right)}\right)$, and $\gamma$ is chosen as the minimum of (8) then with probability at least $1 - \beta$ the error converges to a neighborhood of the global optimum of size*

$$
\min_{t \in [0,K]} f(x^t) - f(x^\star) \leq \mathcal{O}\left(\max\left\{(11), (12), (13), (14)\right\}\right). \tag{10}
$$

*where*

$$\frac{LR^2}{K} + \frac{L^3R^4}{\lambda^2 K^2} \tag{11}$$

$$R\lambda^{1-\alpha/2}\sigma^{\alpha/2}\sqrt{\frac{\ln K/\beta}{K}} + \frac{LR^2\sigma^\alpha \ln K/\beta}{K} \tag{12}$$

$$\frac{R(\sigma^\alpha+\zeta_\lambda^\alpha)\left(\frac{LR}{\lambda}+\frac{\lambda^{\alpha-1}\zeta_\lambda}{\sigma^\alpha+\zeta_\lambda^\alpha}+(\sigma^\alpha+\zeta_\lambda^\alpha)^{\frac{-1}{\alpha}}\right)}{\lambda^{\alpha-1}} + \frac{R^2L(\sigma^\alpha+\zeta_\lambda^\alpha)^2\left(\frac{LR}{\lambda}+\frac{\lambda^{\alpha-1}\zeta_\lambda}{\sigma^\alpha+\zeta_\lambda^\alpha}+(\sigma^\alpha+\zeta_\lambda^\alpha)^{\frac{-1}{\alpha}}\right)^2}{\lambda^{2\alpha}} \tag{13}$$

$$\frac{R\lambda}{\varepsilon}\sqrt{d\ln\left(\frac{K}{\beta}\right)\ln\left(\frac{K}{\delta}\right)\ln\left(\frac{1}{\delta}\right)} + \frac{LR^2 d\ln\left(\frac{K}{\beta}\right)\ln\left(\frac{K}{\delta}\right)\ln\left(\frac{1}{\delta}\right)}{\varepsilon^2}. \tag{14}$$

One may notice that there is a non-trivial trade-off between the convergence rate, clipping level, and the size of the neighborhood. Therefore, we consider two special cases and provide the result with optimally selected $\lambda$ in the following corollary.

**Corollary 4.3** (Convergence of DP-Clipped-SGD for the convex objective)**.** *Let the assumptions of Theorem 4.1 hold, $K$ is sufficiently large, $\gamma$ is chosen as the minimum of (8), $\sigma_\omega = \Theta\left(\frac{\lambda}{\varepsilon}\sqrt{K\ln\left(\frac{K}{\delta}\right)\ln\left(\frac{1}{\delta}\right)}\right)$, and the $\lambda > 4LR$. Then the optimal value for $\lambda$ is*

$$\lambda = \max\left\{4LR, \left(\frac{\varepsilon\sigma^\alpha}{d\ln\left(\frac{K}{\delta}\right)\ln\left(\frac{1}{\delta}\right)\ln\frac{K}{\beta}}\right)^{\frac{1}{\alpha}}\right\}.$$

*With this value, the iterates produced by the algorithm with probability of at least $1-\beta$ satisfy*

$$\min_{k\in[0,K]} f(x^t) - f(x^\star) = \mathcal{O}\left(\max\left\{(15),(16),(17),(18)\right\}\right),$$

*where*

$$\max\left\{\sqrt{\frac{R^{4-\alpha}L^{2-\alpha}\sigma^\alpha\ln\left(\frac{K}{\beta}\right)}{K}}, R\left(\frac{\varepsilon\sigma^\alpha}{\sqrt{d\ln\left(\frac{K}{\delta}\right)\ln\left(\frac{1}{\delta}\right)}}\right)^{\frac{1}{\alpha}}\sqrt{\frac{\ln^{\frac{3\alpha-2}{2\alpha}}\left(\frac{K}{\beta}\right)}{K}}\right\} \tag{15}$$

$$\min\left\{\frac{R^{2-\alpha}\sigma^\alpha}{L^{\alpha-1}}, R\sigma\left(\frac{\sqrt{d\ln\left(\frac{K}{\delta}\right)\ln\left(\frac{1}{\delta}\right)}}{\varepsilon}\right)^{\frac{\alpha-1}{\alpha}}\right\} \tag{16}$$

$$\min\left\{\frac{LR^2}{K^2}, \frac{L^3R^4\left(d\ln\left(\frac{K}{\delta}\right)\ln\left(\frac{1}{\delta}\right)\ln\left(\frac{K}{\beta}\right)\right)^{\frac{1}{\alpha}}}{(\varepsilon)^{\frac{1}{\alpha}}\sigma K^2}\right\} + \frac{LR^2}{K} \tag{17}$$

$$\max\left\{\frac{LR^2}{\varepsilon}\sqrt{d\ln\left(\frac{K}{\delta}\right)\ln\left(\frac{1}{\delta}\right)\ln\left(\frac{K}{\beta}\right)}, \frac{R\sigma\left(d\ln\left(\frac{K}{\delta}\right)\ln\left(\frac{1}{\delta}\right)\ln\left(\frac{K}{\beta}\right)\right)^{\frac{\alpha+2}{2\alpha}}}{\varepsilon^{\frac{\alpha-1}{\alpha}}}\right\}$$

$$+\frac{LR^2 d}{\varepsilon^2}\ln\left(\frac{K}{\delta}\right)\ln\left(\frac{1}{\delta}\right)\ln\left(\frac{K}{\beta}\right). \tag{18}$$

*Also, for small $\lambda$ regime $\left(\lambda \leq \frac{4}{3}LR\right)$, the optimal value for $\lambda$ is*

$$\lambda = \min\left\{\frac{4}{3}LR, \frac{2\varepsilon LR}{\left(d\ln\left(\frac{K}{\delta}\right)\ln\left(\frac{1}{\delta}\right)\ln\frac{K}{\beta}\right)^{\frac{1}{2\alpha+2}}+1}\right\}. \tag{19}$$

*With this value, the iterates produced by the algorithm with probability of at least $1-\beta$ satisfy*

$$\min_{t\in[0,K]} f(x^t) - f(x^\star) = \mathcal{O}\left(\max\left\{(20),(21),(22),(23)\right\}\right),$$

Table 1: Rate, neighborhood and optimal $\lambda$ in different regimes for the convex objective function. Here, $\lambda$ denotes the clipping level, $L$ denotes the smoothness parameter, $R \geq \|x^0 - x^*\|$ represents the initial error, $\alpha \in (1, 2]$ denotes the moment that is bounded and $\sigma^\alpha$ is that upper bound value. Furthermore, $\beta$ is the confidence level, $\zeta_\lambda := \max\{0, 2LR - \frac{\lambda}{2}\}$, and $\eta$ is a small positive constant.

| Regime | Neighborhood | Optimal $\lambda$ | Convergence rate | Optimal Neighborhood |
|---|---|---|---|---|
| $\lambda > 4LR$ $(\zeta_\lambda = 0)$ | $\mathcal{O}\left(R\frac{\sigma^\alpha}{\lambda^{\alpha-1}} + LR^2\frac{\sigma^{2\alpha}}{\lambda^{2\alpha}}\right)$ | $\mathcal{O}\left(\sigma\left(\frac{K}{\ln\frac{K}{\beta}}\right)^{\frac{1}{\alpha}}\right)$ | $\mathcal{O}\left(\left(\frac{\ln\frac{K}{\beta}}{K}\right)^{\frac{\alpha-1}{\alpha}} + \frac{\ln^2\frac{K}{\beta}}{K^2}\right)$ | - |
| $\frac{4}{3}LR < \lambda \leq 4LR$ $\zeta_\lambda < \lambda < \sigma$ | $\mathcal{O}\left(R\frac{\sigma^\alpha}{\lambda^{\alpha-1}} + LR^2\frac{\sigma^{2\alpha}}{\lambda^{2\alpha}}\right)$ | $4LR$ | $\mathcal{O}\left(\sqrt{\frac{\ln\frac{K}{\beta}}{K}} + \frac{\ln\frac{K}{\beta}}{K}\right)$ | $\mathcal{O}\left(\frac{R^{2-\alpha}\sigma^\alpha}{L^{\alpha-1}} + \frac{\sigma^{2\alpha}}{L^{2\alpha-1}R^{2\alpha-2}}\right)$ |
| $\frac{4}{3}LR < \lambda \leq 4LR$ $\zeta_\lambda < \sigma < \lambda$ | $\mathcal{O}\left(R\frac{\sigma^\alpha}{\lambda^{\alpha-1}} + LR^2\frac{\sigma^{2\alpha}}{\lambda^{2\alpha}}\right)$ | $4LR$ | $\mathcal{O}\left(\sqrt{\frac{\ln\frac{K}{\beta}}{K}} + \frac{\ln\frac{K}{\beta}}{K}\right)$ | $\mathcal{O}\left(\frac{R^{2-\alpha}\sigma^\alpha}{L^{\alpha-1}} + \frac{\sigma^{2\alpha}}{L^{2\alpha-1}R^{2\alpha-2}}\right)$ |
|  | $\mathcal{O}\left(R\zeta_\lambda + \frac{LR^2\zeta_\lambda^2}{\lambda^2}\right)$ | $4LR - \eta$ | $\mathcal{O}\left(\sqrt{\frac{\ln\frac{K}{\beta}}{K}} + \frac{\ln\frac{K}{\beta}}{K}\right)$ | $\mathcal{O}\left(R\eta + \frac{LR^2\eta^2}{(LR-\eta)^2}\right)$ |
| $\frac{4}{3}LR < \lambda \leq 4LR$ $(\sigma < \zeta_\lambda < \lambda)$ | $\mathcal{O}\left(R\zeta_\lambda + \frac{LR^2\zeta_\lambda^2}{\lambda^2}\right)$ | $4LR - 2\sigma$ | $\mathcal{O}\left(\sqrt{\frac{\ln\frac{K}{\beta}}{K}} + \frac{\ln\frac{K}{\beta}}{K}\right)$ | $\mathcal{O}\left(R\sigma + \frac{LR^2\sigma^2}{(LR-\sigma)^2}\right)$ |
| $\lambda \leq \frac{4}{3}LR$ $(\lambda < \zeta_\lambda < \sigma)$ | $\mathcal{O}\left(R\frac{\sigma^\alpha\zeta_\lambda}{\lambda^\alpha} + \frac{LR^2\sigma^{2\alpha}\zeta_\lambda^2}{\lambda^{2\alpha+2}}\right)$ | $\frac{4}{3}LR$ | $\mathcal{O}\left(\sqrt{\frac{\ln\frac{K}{\beta}}{K}} + \frac{\ln\frac{K}{\beta}}{K}\right)$ | $\mathcal{O}\left(\frac{R^{2-\alpha}\sigma^\alpha}{L^{\alpha-1}} + \frac{\sigma^{2\alpha}}{L^{2\alpha-1}R^{2\alpha-2}}\right)$ |
| $\lambda \leq \frac{4}{3}LR$ $(\lambda < \sigma < \zeta_\lambda)$ | $\mathcal{O}\left(R\frac{\zeta_\lambda^{\alpha+1}}{\lambda^\alpha} + \frac{LR^2\zeta_\lambda^{2\alpha}}{\lambda^{2\alpha+2}}\right)$ | $\frac{4}{3}LR - \eta$ | $\mathcal{O}\left(\sqrt{\frac{\ln\frac{K}{\beta}}{K}} + \frac{\ln\frac{K}{\beta}}{K}\right)$ | $\mathcal{O}\left(\frac{R(LR+\eta)^{\alpha+1}}{(LR-\eta)^\alpha} + \frac{LR^2(LR+\eta)^{2\alpha}}{(LR-\eta)^{2\alpha+2}}\right)$ |
|  | $\mathcal{O}\left(R\frac{\zeta_\lambda^{\alpha+1}}{\lambda^\alpha} + \frac{LR^2\zeta_\lambda^{2\alpha}}{\lambda^{2\alpha+2}}\right)$ | $\frac{4}{3}LR - \eta$ | $\mathcal{O}\left(\sqrt{\frac{\ln\frac{K}{\beta}}{K}} + \frac{\ln\frac{K}{\beta}}{K}\right)$ | $\mathcal{O}\left(\frac{R(LR+\eta)^{\alpha+1}}{(LR-\eta)^\alpha} + \frac{LR^2(LR+\eta)^{2\alpha}}{(LR-\eta)^{2\alpha+2}}\right)$ |
| $\lambda \leq \frac{4}{3}LR$ $(\sigma < \lambda < \zeta_\lambda)$ | $\mathcal{O}\left(R\frac{\sigma^{\alpha-1}}{\lambda^{\alpha-1}} + \frac{LR^2\sigma^2\zeta_\lambda^{2\alpha-2}}{\lambda^{2\alpha}}\right)$ | $\frac{4}{3}LR$ | $\mathcal{O}\left(\sqrt{\frac{\ln\frac{K}{\beta}}{K}} + \frac{\ln\frac{K}{\beta}}{K}\right)$ | $\mathcal{O}\left(R\sigma + \frac{\sigma^2}{L}\right)$ |

*where*

$$\min\left\{\sqrt{\frac{R^{4-\alpha}L^{2-\alpha}\sigma^\alpha \ln\left(\frac{K}{\beta}\right)}{K}}, \sqrt{\frac{R^{4-\alpha}(\varepsilon L)^{2-\alpha} \ln^{\frac{3\alpha}{4\alpha+4}}\left(\frac{K}{\beta}\right)}{\left(d\ln\left(\frac{K}{\delta}\right)\ln\left(\frac{1}{\delta}\right)\right)^{\frac{2-\alpha}{4\alpha+4}} K}}\right\} \tag{20}$$

$$\max\left\{\frac{R^{2-\alpha}\sigma^\alpha}{L^{\alpha-1}}, \frac{R^{2-\alpha}\sigma^\alpha}{\varepsilon}\left(d\ln\left(\frac{K}{\delta}\right)\ln\left(\frac{1}{\delta}\right)\ln\left(\frac{K}{\beta}\right)\right)^{\frac{\alpha-1}{2\alpha+2}}\right\} \tag{21}$$

$$\max\left\{\frac{LR^2}{K^2}, \frac{LR^2}{\varepsilon^2 K^2}\left(d\ln\left(\frac{K}{\delta}\right)\ln\left(\frac{1}{\delta}\right)\ln\left(\frac{K}{\beta}\right)\right)^{\frac{2}{2\alpha+2}}\right\} + \frac{LR^2}{K} \tag{22}$$

$$\min\left\{\frac{LR^2}{\varepsilon}\sqrt{d\ln\left(\frac{K}{\delta}\right)\ln\left(\frac{1}{\delta}\right)\ln\left(\frac{K}{\beta}\right)}, \frac{LR^2}{\left(d\ln\left(\frac{K}{\delta}\right)\ln\left(\frac{1}{\delta}\right)\ln\left(\frac{K}{\beta}\right)\right)^{\frac{1}{2\alpha+2}}}\right\}$$
$$+ \frac{LR^2 d}{\varepsilon^2}\ln\left(\frac{K}{\delta}\right)\ln\left(\frac{1}{\delta}\right)\ln\left(\frac{K}{\beta}\right). \tag{23}$$

In the finite-sum case, i.e., when $f(x) = \frac{1}{n}\sum_{i=1}^n f_i(x)$ for some finite $n$, Abadi et al. (2016) show that it is sufficient to choose $\sigma_\omega = \Theta\left(\frac{q\lambda}{\varepsilon}\sqrt{K\ln\frac{1}{\delta}}\right)$, where $q = b/n$, $b$ is the mini-batch size, clipping is applied to each stochastic gradient, and $\varepsilon = \mathcal{O}(q^2 K)$, allowing to have smaller $\varepsilon$ and $\delta$ for given $\sigma_\omega$ and $\lambda$. We note that our analysis holds for the finite-sum case without changes as long as the assumptions of the theorem are satisfied and the mini-batch size equals 1.

**Non-convex problems.** In the non-convex case, we derive the following result.

**Theorem 4.4** (Convergence of DP-Clipped-SGD for the non-convex objective). *Let the integer $K \geq 0$ and $\beta \in (0, 1]$ be given. Let the assumptions 2.1, 2.2, 2.4, hold for the set $Q$ defined as $Q = \left\{x \in \mathbb{R} \mid \exists\, y \in \mathbb{R}^d : f(y) \leq f^* + 2\Delta \text{ and } \|x - y\| \leq \sqrt{\Delta}/20\sqrt{L}\right\}$, where $\Delta \geq f(x^0) - f^*$,*

Table 2: Rate, neighborhood and optimal $\lambda$ in different regimes for the non-convex objective function. Here, $\lambda$ denotes the clipping level, $L$ denotes the smoothness parameter, $\Delta \geq f(x^0) - f(x^*)$ represents the initial error, $\alpha \in (1, 2]$ denotes the moment that is bounded and $\sigma^\alpha$ is that upper bound value. Furthermore, $\beta$ is the confidence level, $\zeta_\lambda := \max\{0, 2\sqrt{L\Delta} - \frac{\lambda}{2}\}$, and $\eta$ is a small positive constant.

| Regime | Neighborhood | Optimal $\lambda$ | Convergence rate | Optimal Neighborhood |
|---|---|---|---|---|
| $\lambda > 4\sqrt{L\Delta}$ $(\zeta_\lambda = 0)$ | $\mathcal{O}\left(\sqrt{L\Delta}\frac{\sigma^\alpha}{\lambda^{\alpha-1}} + L\Delta\frac{\sigma^{2\alpha}}{\lambda^{2\alpha}}\right)$ | $\mathcal{O}\left(\sigma\left(\frac{K}{\ln\frac{K}{\beta}}\right)^{\frac{1}{\alpha}}\right)$ | $\mathcal{O}\left(\left(\frac{\ln\frac{K}{\beta}}{K}\right)^{\frac{\alpha-1}{\alpha}} + \frac{\ln^2\frac{K}{\beta}}{K^2}\right)$ | - |
| $\frac{4}{3}\sqrt{L\Delta} < \lambda \leq 4\sqrt{L\Delta}$ $\zeta_\lambda < \lambda < \sigma$ | $\mathcal{O}\left(\sqrt{L\Delta}\frac{\sigma^\alpha}{\lambda^{\alpha-1}} + L\Delta\frac{\sigma^{2\alpha}}{\lambda^{2\alpha}}\right)$ | $4\sqrt{L\Delta}$ | $\mathcal{O}\left(\sqrt{\frac{\ln\frac{K}{\beta}}{K}} + \frac{\ln\frac{K}{\beta}}{K}\right)$ | $\mathcal{O}\left(\frac{\sigma^\alpha}{(\sqrt{L\Delta})^{\alpha-2}} + \frac{\sigma^{2\alpha}}{(L\Delta)^{2\alpha-4}}\right)$ |
| $\frac{4}{3}\sqrt{L\Delta} < \lambda \leq 4\sqrt{L\Delta}$ $\zeta_\lambda < \lambda < \sigma$ | $\mathcal{O}\left(\sqrt{L\Delta}\frac{\sigma^\alpha}{\lambda^{\alpha-1}} + L\Delta\frac{\sigma^{2\alpha}}{\lambda^{2\alpha}}\right)$ | $4\sqrt{L\Delta}$ | $\mathcal{O}\left(\sqrt{\frac{\ln\frac{K}{\beta}}{K}} + \frac{\ln\frac{K}{\beta}}{K}\right)$ | $\mathcal{O}\left(\frac{\sigma^\alpha}{(\sqrt{L\Delta})^{\alpha-2}} + \frac{\sigma^{2\alpha}}{(L\Delta)^{2\alpha-4}}\right)$ |
| | $\mathcal{O}\left(\sqrt{L\Delta}\zeta_\lambda + \frac{L\Delta\zeta_\lambda^2}{\lambda^2}\right)$ | $4\sqrt{L\Delta} - \eta$ | $\mathcal{O}\left(\sqrt{\frac{\ln\frac{K}{\beta}}{K}} + \frac{\ln\frac{K}{\beta}}{K}\right)$ | $\mathcal{O}\left(\sqrt{L\Delta}\eta + \frac{L\Delta\eta^2}{(\sqrt{L\Delta}-\eta)^2}\right)$ |
| $\frac{4}{3}\sqrt{L\Delta} < \lambda \leq 4\sqrt{L\Delta}$ $(\sigma < \zeta_\lambda < \lambda)$ | $\mathcal{O}\left(\sqrt{L\Delta}\zeta_\lambda + \frac{L\Delta\zeta_\lambda^2}{\lambda^2}\right)$ | $4\sqrt{L\Delta} - 2\sigma$ | $\mathcal{O}\left(\sqrt{\frac{\ln\frac{K}{\beta}}{K}} + \frac{\ln\frac{K}{\beta}}{K}\right)$ | $\mathcal{O}\left(\sqrt{L\Delta}\sigma + \frac{L\Delta\sigma^2}{(\sqrt{L\Delta}-\sigma)^2}\right)$ |
| $\lambda \leq \frac{4}{3}\sqrt{L\Delta}$ $(\lambda < \zeta_\lambda < \sigma)$ | $\mathcal{O}\left(\sqrt{L\Delta}\frac{\sigma^\alpha\zeta_\lambda}{\lambda^\alpha} + \frac{L\Delta\sigma^{2\alpha}\zeta_\lambda^2}{\lambda^{2\alpha+2}}\right)$ | $\frac{4}{3}\sqrt{L\Delta}$ | $\mathcal{O}\left(\sqrt{\frac{\ln\frac{K}{\beta}}{K}} + \frac{\ln\frac{K}{\beta}}{K}\right)$ | $\mathcal{O}\left(\frac{\sigma^\alpha}{(\sqrt{L\Delta})^{\alpha-2}} + \frac{\sigma^{2\alpha}}{(L\Delta)^{2\alpha-4}}\right)$ |
| $\lambda \leq \frac{4}{3}\sqrt{L\Delta}$ $(\lambda < \sigma < \zeta_\lambda)$ | $\mathcal{O}\left(\sqrt{L\Delta}\frac{\zeta_\lambda^{\alpha+1}}{\lambda^\alpha} + \frac{L\Delta\zeta_\lambda^{2\alpha}}{\lambda^{2\alpha+2}}\right)$ | $\frac{4}{3}\sqrt{L\Delta} - \eta$ | $\mathcal{O}\left(\sqrt{\frac{\ln\frac{K}{\beta}}{K}} + \frac{\ln\frac{K}{\beta}}{K}\right)$ | $\mathcal{O}\left(\frac{\sqrt{L\Delta}(\sqrt{L\Delta}+\eta)^{\alpha+1}}{(\sqrt{L\Delta}-\eta)^\alpha} + \frac{L\Delta(\sqrt{L\Delta}+\eta)^{2\alpha}}{(\sqrt{L\Delta}-\eta)^{2\alpha+2}}\right)$ |
| | $\mathcal{O}\left(\sqrt{L\Delta}\frac{\zeta_\lambda^{\alpha+1}}{\lambda^\alpha} + \frac{L\Delta\zeta_\lambda^{2\alpha+2}}{\lambda^{2\alpha+2}}\right)$ | $\frac{4}{3}\sqrt{L\Delta} - \eta$ | $\mathcal{O}\left(\sqrt{\frac{\ln\frac{K}{\beta}}{K}} + \frac{\ln\frac{K}{\beta}}{K}\right)$ | $\mathcal{O}\left(\frac{\sqrt{L\Delta}(\sqrt{L\Delta}+\eta)^{\alpha+1}}{(\sqrt{L\Delta}-\eta)^\alpha} + \frac{L\Delta(\sqrt{L\Delta}+\eta)^{2\alpha}}{(\sqrt{L\Delta}-\eta)^{2\alpha+2}}\right)$ |
| $\lambda \leq \frac{4}{3} \cdot 4\sqrt{L\Delta}$ $(\sigma < \lambda < \zeta_\lambda)$ | $\mathcal{O}\left(\sqrt{L\Delta}\frac{\sigma\zeta_\lambda^{\alpha-1}}{\lambda^{\alpha-1}} + L\Delta\frac{\sigma^2\zeta_\lambda^{2\alpha-2}}{\lambda^{2\alpha}}\right)$ | $\frac{4}{3}\sqrt{L\Delta}$ | $\mathcal{O}\left(\sqrt{\frac{\ln\frac{K}{\beta}}{K}} + \frac{\ln\frac{K}{\beta}}{K}\right)$ | $\mathcal{O}\left(\sqrt{L\Delta}\sigma + \sigma^2\right)$ |

$\zeta_\lambda := \max\left\{0, 2\sqrt{L\Delta} - \frac{\lambda}{2}\right\}$, and $\gamma$ is selected according to

$$\gamma \leq \mathcal{O}\left(\min\left\{\frac{1}{L}, \frac{\sqrt{\frac{\Delta}{L}}}{\lambda^{1-\alpha/2}\sqrt{K\ln\left(\frac{K}{\beta}\right)(\sigma^\alpha + \zeta_\lambda^\alpha)}}\right.\right.,$$

$$\left.\left.\frac{\sqrt{\frac{\Delta}{L}}\lambda^{\alpha-1}}{K(\sigma^\alpha + \zeta_\lambda^\alpha)\left(\frac{\sqrt{L\Delta}}{\lambda} + \frac{\lambda^{\alpha-1}\zeta_\lambda}{\sigma^\alpha+\zeta_\lambda^\alpha} + (\sigma^\alpha + \zeta_\lambda^\alpha)^{\frac{-1}{\alpha}}\right)}, \frac{\sqrt{\frac{\Delta}{L}}}{\sigma_\omega\sqrt{dK\ln\left(\frac{K}{\beta}\right)}}\right\}\right). \tag{24}$$

*Then, after $K$ iterations of DP-Clipped-SGD and with probability at least $1 - \beta$, we have*

$$\min_{t\in[0,K]}\left\|\nabla f(x^t)\right\|^2 \leq \frac{8\Delta}{\gamma(K+1)} + \frac{128\Delta^2}{\lambda^2\gamma^2(K+1)^2} \tag{25}$$

Similarly to the convex case, the above result establishes the convergence to a certain neighborhood with a faster $\mathcal{O}(1/\sqrt{K})$ rate. We defer the corollaries for the non-convex case to the appendix and describe different special cases for the no-DP regime in Table 2.

*Proof sketch.* The proof of Theorems 4.1 and 4.4 is heavily inspired by (Sadiev et al., 2023). Yet, there is a crucial difference in defining the clipping level parameter. In contrast to (Sadiev et al., 2023), we treat $\lambda$ as given rather than calculating it based on other problem parameters. By doing so, the fundamental assumption regarding the magnitude of $\lambda$ in comparison to the norm of the gradient in bias-variance of the clipped vector (Lemma 5.1) of (Sadiev et al., 2023) becomes invalid. Thus, we develop a general bias-variance lemma (Lemma B.1) to study the statistical properties of the clipped vector.

# 5 Conclusion

In this paper, we present the first high-probability convergence analysis of DP-Clipped-SGD for both convex and non-convex smooth optimization problems under heavy-tailed noise. Our results demonstrate that DP-Clipped-SGD converges to a certain neighborhood of the optimal solution at a rate of $\mathcal{O}(1/\sqrt{K})$. In future work, it would be valuable to extend these results to the Federated Learning setting and to investigate the tightness and optimality of the derived bounds.

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
