# OpenReview forum: "Differentially Private Clipped-SGD: High-Probability Convergence with Arbitrary Clipping Level"
_NeurIPS.cc/2025/Conference — Submitted to NeurIPS 2025_

### Official Review · Reviewer_5QfS · 2025-06-23

**Clarity:** 4
**Significance:** 4
**Originality:** 4
**Rating:** 5
**Confidence:** 3

**Summary:**

This paper presents a high-probability convergence analysis of differentially private clipped SGD and establishes a convergence neighborhood that, in certain regimes, closely matches the expected value. It addresses the challenge of high-variance noise introduced by differential privacy in gradient estimators and introduces techniques to handle arbitrary, time-invariant clipping levels, providing an analysis that effectively manages this noise for DP while maintaining favorable high-probability bounds.

**Questions:**

None

**Ethical Concerns:**

["NO or VERY MINOR ethics concerns only"]

**Final Justification:**

I'm happy to keep my original score, as I didn’t have major concerns with the technical content or the main results of the paper. My main suggestion was to improve the clarity and readability, especially for readers who are new to high-probability analysis in optimization.
The authors have said they’ll make those changes, and I think the contribution is solid enough for the paper to be accepted. That said, since the technical analysis mostly builds on existing techniques, I don’t see a strong reason to increase my score either.

**Limitations:**

Yes

**Quality:**

4

**Strengths And Weaknesses:**

**Strengths:**

1. The paper presents a well-executed theoretical analysis that helps clarify the behavior of differentially private SGD, particularly in high-probability settings.
2. It offers useful insights into handling time-invariant clipping and high-variance estimators—two major challenges for obtaining strong high-probability bounds in DP optimization.
3. The related work is well summarized, and the paper effectively builds on and combines ideas from prior work to reach cleaner and more interpretable results.
4. The authors clearly point out the limitations of existing approaches and do a good job of situating their contributions within the broader context, especially through intuitive proof sketches.

**Weaknesses:**

1. Some of the theoretical results are quite dense, and the paper would benefit from an informal summary of the key takeaways—especially the regimes where the results offer improvements over prior work. This would make it easier for readers to interpret the significance of the

---

> ### Author Rebuttal · Authors · 2025-07-28
>
> We sincerely thank the reviewer for the thoughtful and highly positive evaluation of our work. We are especially grateful for the recognition of the significance, originality, and clarity of the theoretical contributions, as well as the emphasis on how the work advances the understanding of DP-Clipped-SGD in high-probability settings. Below, we address your helpful suggestion.
>
> > **Comment**: Some of the theoretical results are quite dense, and the paper would benefit from an informal summary of the key takeaways — especially the regimes where the results offer improvements over prior work.
>
> **Response**:  Thank you for this excellent suggestion. We fully agree that providing an informal summary would enhance the accessibility of our results. In the final version of the paper, we will add:
>
> - A concise and intuitive summary at the end of Section 4 (right after new paragraph about the detailed comparison with Koloskova et al. (2023) that we provided in our response to Reviewer hmsX), highlighting the key insights behind the convergence bounds in both the convex and non-convex settings.
>
> - Clear interpretations of how our results compare to prior work under different clipping and privacy regimes (e.g., fixed vs. growing clipping level, zero vs. non-zero DP noise).

---

> ### Comment · Reviewer_5QfS · 2025-08-02
>
> Many thanks to the authors for kindly addressing all my concerns! I’m happy with the current score and would like to keep it as is.

---

> > ### Author Response · Authors · 2025-08-05
> >
> > Thank you very much for your kind message and for your thoughtful and constructive review. We truly appreciate your engagement throughout the process and are glad to hear that our responses addressed your concerns. We’re grateful for your support and are pleased with your positive evaluation of our work.

---

### Official Review · Reviewer_hmsX · 2025-07-02

**Clarity:** 1
**Significance:** 2
**Originality:** 2
**Rating:** 2
**Confidence:** 4

**Summary:**

The paper studies noisy clipped SGD algorithm applied to differential private stochastic optimization, for convex and non-convex objectives, under heavy-tailed noises and arbitrary fixed clipping level. Assuming smooth objective function and $\alpha$-th moment bound on the stochastic gradient noise for some $\alpha \in (1, 2]$, the paper shows a collection of optimization error bounds:
1. A general result for noisy clipped SGD in the convex case.
2. By tuning the noisy variance to ensure differential privacy and choosing the clipping level optimally, the paper shows convergence to a level set of the objective function, with a bounded function value gap.
3. Similar bounds are derived for the gradient norms in non-convex optimization.

**Questions:**

In order for the paper to be publishable, the authors need to address a basic question: is it possible at all to establish convergence results (i.e. the bound goes to 0 as K goes to infinity) in the heavy-tailed DP-clipped-SGD settings? The results could be either for function value in convex case, or gradient norm in non-convex case, but a non-vacuous bound must at least show a convergence rate. Through more careful arguments by analyzing the distribution of clipped stochastic gradients, this might be possible.

**Ethical Concerns:**

["NO or VERY MINOR ethics concerns only"]

**Final Justification:**

After discussing with authors during rebuttal period, I better understand the potential applicability to finite-sum settings, with non-vacuous bounds. However, these bounds are never formally stated or proved, and the dependence on sample size (as well as parameter choices w.r.t. sample size) are not clear.

Consequently, there might be potentially an interesting result hidden here, but the way the authors choose to present is so poor that the main results look just like vacuous bounds. I therefore decide to keep my rating, with encouragement for the authors to re-write the result highlighting non-vacuous bounds.

**Limitations:**

The theoretical results in the paper have limitations due to the non-convergence. The paper does not involve any potential negative societal impacts.

**Paper Formatting Concerns:**

No formatting concerns.

**Quality:**

2

**Strengths And Weaknesses:**

Strength:
- The paper deals with challenging heavy-tailed cases where the second moment of gradient noise does not exist.
- Compared to existing literature, the results address the choice of clipping level based on differential privacy requirements.

Weakness:
The major weakness of the paper is that the final bounds do not decrease with the number of iterations. If we see the smoothness parameters and the differential privacy parameter as constants, the upper bounds for private learning are of constant order, and therefore vacuous. Under the assumptions in the paper, it is not even clear whether these bounds are better than the quality of the initial point $x_0$, as we have $f (x_0) - f (x^*) \leq L/2 R^2/2$, and a trivial algorithm that outputs $x_0$ is differentially private.

The bounds in DP results (e.g. Corollary 4.3) are generally difficult to parse. It is better to simplify the bound under appropriate sample size conditions and parameter selections, to see the magnitude of the bounds w.r.t. problem parameters.

---

> ### Author Rebuttal · Authors · 2025-07-28
>
> We thank the reviewer for the thoughtful review and helpful suggestions. We appreciate your recognition of the significance of our contribution in handling heavy-tailed noise with a fixed clipping threshold under differential privacy (DP). Below, we address your concerns and suggestions in detail.
>
> > **Q1**: The major weakness of the paper is that the final bounds do not decrease with the number of iterations. If we see the smoothness parameters and the differential privacy parameter as constants, the upper bounds for private learning are of constant order, and therefore vacuous.
>
> **A1**: Thank you for raising this important point. Indeed, our high-probability results in the DP setting show convergence to a non-vanishing neighborhood — a common (Koloskova et al., 2023) and even unavoidable (Allouah et al., 2023) phenomenon in differentially private optimization. This neighborhood size depends on the fixed clipping threshold $\lambda$ and the magnitude of DP noise $\sigma_\omega$, which itself scales as $\sigma_\omega = \mathcal{O}\left(\frac{\lambda}{\epsilon} \sqrt{K \ln(K/\delta) \ln(1/\delta)}\right)$.
>
> Moreover, even in the non-DP setting, i.e., when $\sigma_\omega = 0$, our high-probability bounds imply that Clipped-SGD with fixed clipping level $\lambda$ also converges only to some neighborhood depending on $\lambda$. From this perspective, our bound is aligned with the lower bound from Koloskova et al. (2023), yet there are certain differences that we discuss at the end of our reply.
>
> > **Q2**: Under the assumptions in the paper, it is not even clear whether these bounds are better than the quality of the initial point $x_0$, as we have $f(x_0) - f(x^\ast) \leq L / 2R^2 / 2$, and a trivial algorithm that outputs $x_0$ is differentially private.
>
> **A2**: Thank you for this important question. Our results indeed imply a nontrivial improvement over the initial point $x_0$ when the step size is appropriately chosen to balance optimization and privacy noise. As one can see from Corollary 4.2, the terms from (11) and (12) are decreasing in $K$. Next, the terms from (13) can be decreased by increasing $\lambda$ or by decreasing $\sigma$, which can be achieved using mini-batching even for the heavy-tailed noise (Kornilov et al., 2023). Finally, the DP-neighborhood terms from (14) can be made smaller via sacrificing the privacy, i.e., via an increase of $\epsilon$.
>
> We will add this discussion in the final version.
>
> > **Q3**: The bounds in DP results (e.g. Corollary 4.3) are generally difficult to parse. It is better to simplify the bound under appropriate sample size conditions and parameter selections, to see the magnitude of the bounds w.r.t. problem parameters.
>
> **A3**: Thank you for this suggestion. We agree that the current presentation of Corollary 4.3 could be more digestible. In the final version, we will provide simplified expressions under different conditions on $\epsilon$ (“small” and “large” $\epsilon$) and $\sigma$. This should help make the implications of our results more transparent.
>
> > **Q4**: In order for the paper to be publishable, the authors need to address a basic question: is it possible at all to establish convergence results (i.e. the bound goes to 0 as K goes to infinity) in the heavy-tailed DP-clipped-SGD settings?
>
> **A4**: This is a deep and important question. In our setting, achieving convergence to zero requires either:
>
> Increasing the clipping threshold $\lambda$ with $K$ — which breaks compatibility with standard DP mechanisms, or
>
> Decreasing the noise magnitude $\sigma_\omega$ with $K$ — which would compromise the $(\epsilon, \delta)$-DP guarantee.
>
> Under fixed clipping and fixed privacy parameters, convergence to the exact optimum is provably impossible (Allouah et al., 2023; Koloskova et al., 2023), and this is inherent to the privacy-utility tradeoff. However, if one relaxes privacy slightly, it may be possible to derive bounds that decay with $K$ under milder assumptions.
>
> We will expand the discussion in the limitations section to highlight these theoretical challenges and explicitly mention this impossibility under fixed $\lambda$ and fixed DP parameters.
>
> ## Comparison with the results by Koloskova et al. (2023)
>
> Motivated by the questions raised by the reviewer, we plan to extend the discussion of the obtained results in the final version of our paper. In particular, we promise to include the following text comparing our results in the non-convex case with closely related ones from Koloskova et al. (2023).
>
> We start with the corollary of Theorem 4.4.
>
> **Corollary 4.5.** Let the assumption of Theorem 4.4 hold, and  $\gamma$ is chosen as the minimum of (24). Then, with probability at least $1-\beta$
>
> $$\min_{t\in [0,K]} \| \| \nabla f(x^t) \| \|^2 \leq \mathcal{O}\left(\max\left\lbrace (27), (28), (29), (30) \right\rbrace\right),$$
>
> where
>
> $$\frac{L\Delta}{K} + \frac{L^2\Delta^2}{\lambda^2 K^2} \tag{27} $$
> $$\sqrt{L\Delta}\lambda^{1 - \alpha/2}\sqrt{\frac{(\sigma^\alpha + \zeta_\lambda^\alpha)\ln(K/\beta)}{K}} + \frac{L\Delta(\sigma^\alpha + \zeta_\lambda^\alpha)\ln(K/\beta)}{\lambda^\alpha K} \tag{28}$$
> $$\frac{\sqrt{\Delta L}(\sigma^\alpha+\zeta_\lambda^\alpha)\left(\frac{\sqrt{L\Delta}}{\lambda}+\frac{\lambda^{\alpha-1}\zeta_\lambda}{\sigma^\alpha+\zeta_\lambda^\alpha}+\left(\sigma^\alpha+\zeta_\lambda^\alpha\right)^{\frac{-1}{\alpha}}\right)}{\lambda^{\alpha-1}}+\frac{\Delta L(\sigma^\alpha+\zeta_\lambda^\alpha)^2\left(\frac{\sqrt{L\Delta}}{\lambda}+\frac{\lambda^{\alpha-1}\zeta_\lambda}{\sigma^\alpha+\zeta_\lambda^\alpha}+\left(\sigma^\alpha+\zeta_\lambda^\alpha\right)^{\frac{-1}{\alpha}}\right)^2}{\lambda^{2\alpha}} \tag{29}$$
> $$\frac{\sigma_\omega\sqrt{dL\Delta \ln(K/\beta)}}{\sqrt{K}} + \frac{\sigma_\omega^2 dL\Delta \ln(K/\beta)}{\lambda^2 K}. \tag{30}$$
>
> Koloskova et al. (2023) derive their *in-expectation* convergence result under the $(L_0,L_1)$-smoothness assumption and the $\sigma^2$-uniformly bounded variance assumption (i.e., Assumption 2.4 with $\alpha = 2$), for DP-Clipped-SGD with mini-batching. For ease of comparison, we consider the special case $L_1 = 0$ and $L_0 = L$, which corresponds to standard $L$-smoothness. Moreover, for simplicity, we assume a mini-batch size of $1$. In this setting, the result from Koloskova et al. (2023, Appendix C.4.2) for DP-Clipped-SGD can be written as follows: if $\gamma \leq 1/9L$, then
>
> $$\min\limits_{t\in [0,K]}\left(\mathbb{E}\left[\|\| \nabla f(x^t) \|\|\right]\right)^2 \leq \mathcal{O}\left(\frac{\Delta}{\gamma K} + \frac{\Delta^2}{\lambda^2\gamma^2K^2} + \gamma L \sigma^2 +  \min\left\lbrace\sigma^2, \frac{\sigma^4}{\lambda^2}\right\rbrace + \gamma L d\sigma_\omega^2 + \frac{\gamma^2 L^2 d^2 \sigma_\omega^4}{\lambda^2} \right).$$
>
> The structure of our bound is quite similar. Specifically, the terms from (27) correspond to the convergence of DP-Clipped-SGD in the noiseless regime ($\sigma = \sigma_\omega = 0$) and match the $\mathcal{O}\left(\frac{\Delta}{\gamma K} + \frac{\Delta^2}{\lambda^2\gamma^2K^2}\right)$ part when $\gamma = \Theta(1/L)$. Next, the terms in (28) serve as analogs of the $\mathcal{O}(\gamma L \sigma^2)$ term. The leading term in (28) matches the $K$-dependence of $\mathcal{O}(\gamma L \sigma^2)$ for $\gamma = \Theta(1/\sqrt{K})$. However, these terms also depend on the clipping level $\lambda$, which arises from our high-probability convergence analysis and the presence of heavy-tailed noise.
>
> The key difference lies in the terms stemming from the inherent bias of Clipped-SGD (Koloskova et al., 2023, Theorems 3.1–3.2) and the DP noise. In our result, these bias terms appear in (29), while the corresponding term in Koloskova et al. (2023) is $\mathcal{O}\left( \min\left\lbrace\sigma^2, \frac{\sigma^4}{\lambda^2}\right\rbrace \right)$. As shown in Table 2, in the special case $\lambda > 4\sqrt{L\Delta}$, the bias terms (i.e., the convergence neighborhood when $\sigma_\omega = 0$) in (29) reduce to $\mathcal{O}\left(\sqrt{L\Delta}\frac{\sigma^\alpha}{\lambda^{\alpha-1}} + L\Delta \frac{\sigma^{2\alpha}}{\lambda^{2\alpha}}\right)$. Assuming $\lambda > \sigma$ for simplicity, the term from Koloskova et al. (2023) becomes $\mathcal{O}\left(\frac{\sigma^4}{\lambda^2}\right)$, which is strictly larger than the second term and strictly smaller than the first term in our bound when $\alpha = 2$. Furthermore, in this regime, both terms in our bound decrease with increasing $\alpha$, suggesting that the convergence neighborhood grows with the heaviness of the noise. Whether the bound in (29) is tight and whether improvements are possible in other regimes remain open questions.
>
> Finally, ignoring logarithmic factors (introduced by the high-probability analysis), the DP-noise-related terms in our bound (30) are $\tilde{\mathcal{O}}\left(\frac{\sigma_\omega\sqrt{dL\Delta}}{\sqrt{K}} + \frac{\sigma_\omega^2 dL\Delta}{\lambda^2 K}\right)$, while the corresponding terms in Koloskova et al. (2023) are $\mathcal{O}\left(\gamma L d\sigma_\omega^2 + \frac{\gamma^2L^2d^2\sigma_\omega^4}{\lambda^2}\right)$. Setting $\gamma = \sqrt{\Delta/(LdK)}$ yields the latter bound as $\mathcal{O}\left(\frac{\sigma_\omega \sqrt{dL\Delta}}{\sqrt{K}} + \frac{\sigma_\omega^4 d L\Delta}{\lambda^2 K}\right)$, which matches (30) up to logarithmic factors.
>
> ---
>
> References:
>
> Allouah et al. “On the privacy-robustness-utility trilemma in distributed learning”. ICML 2023
>
> Koloskova et al. “Revisiting gradient clipping: Stochastic bias and tight convergence guarantees”. ICML 2023
>
> Kornilov et al. “Accelerated zeroth-order method for non-smooth stochastic convex optimization problem with infinite variance”. NeurIPS 2023

---

> > ### Comment · Reviewer_hmsX · 2025-08-06
> > **Post-rebuttal comments**
> >
> > Thank the authors for writing the response and clarifying comparison to existing works. Thank the authors for being straightforward about the non-vanishing bounds. However, I still think that the non-vanishing bounds make the theoretical contribution weak, and I could not recommend acceptance.
> >
> > In particular,
> >
> > - The bound, for example, contains a term $L R^2 d / \varepsilon^2$. The privacy $\varepsilon$ is usually seen as a constant, while both $L$ and $R$ can grow with problem size (e.g. dimension) and cannot be mitigated by more iterations. If we compute such a quantity on a canonical stochastic convex optimization problem, this usually makes the result vacuous.
> >
> > - I understand author's points that the current analysis framework is faced with difficulties when dealing with private clipped SGD. However, the author only shows a weak upper bound instead of a lower bound. Therefore, the non-vanishing bound may be due to sub-optimal analysis, or sub-optimal choice of parameters in the algorithm.
> >
> > - The author also points out to lower bound by Allouah et al., (2023). That paper establishes lower bounds for private distributed learning with robustness requirements. I don't think Allouah et al., (2023) implies a non-vanishing lower bound for the problem that this paper is considering.

---

> > ### Comment · Reviewer_hmsX · 2025-08-06
> > **Post-rebuttal comments**
> >
> > Thank the authors for writing the response and clarifying comparison to existing works. Thank the authors for being straightforward about the non-vanishing bounds. However, I still think that the non-vanishing bounds make the theoretical contribution weak, and I could not recommend acceptance.
> >
> > In particular,
> >
> > - The bound, for example, contains a term $L R^2 d / \varepsilon^2$. The privacy $\varepsilon$ is usually seen as a constant, while both $L$ and $R$ can grow with problem size (e.g. dimension) and cannot be mitigated by more iterations. If we compute such a quantity on a canonical stochastic convex optimization problem, this usually makes the result vacuous.
> >
> > - I understand author's points that the current analysis framework is faced with difficulties when dealing with private clipped SGD. However, the author only shows a weak upper bound instead of a lower bound. Therefore, the non-vanishing bound may be due to sub-optimal analysis, or sub-optimal choice of parameters in the algorithm.
> >
> > - The author also points out to lower bound by Allouah et al., (2023). That paper establishes lower bounds for private distributed learning with robustness requirements. I don't think Allouah et al., (2023) implies a non-vanishing lower bound for the problem that this paper is considering.

---

> > > ### Author Response · Authors · 2025-08-06
> > >
> > > We sincerely thank the reviewer for the detailed follow-up and for engaging with our response. However, we respectfully disagree with the claims raised in the post-rebuttal comment.
> > >
> > > ## Re 1 & 3: On the bound involving the term $\frac{LR^2d}{\varepsilon^2}$ and the relevance of the lower bound from Allouah et al. (2023)
> > >
> > > Contrary to the reviewer’s assertion, the lower bound from Allouah et al. (2023) **does apply even in non-adversarial settings**. Specifically, Section 3.1 ("Non-adversarial setting") establishes a lower bound of $\Omega\left(\frac{d}{\varepsilon^2 nm^2}\right)$, for distributed problems with $n$ clients, where each client minimizes a convex finite-sum objective over $m$ samples. While the exact setting differs from ours (we consider a more general stochastic optimization problem), the lower bound exhibits the **same dependence on $d$ and $\varepsilon$** as the term $\frac{LR^2d}{\varepsilon^2}$ in our Corollary 4.3.
> > >
> > > Furthermore, in our original rebuttal we provided a detailed comparison of our **non-convex** results with the state-of-the-art upper bounds for DP-Clipped-SGD from Koloskova et al. (2023). As noted, the **DP-induced neighborhood terms in our bounds exactly match those in their analysis**.
> > >
> > >
> > > ## Re 2: "the non-vanishing bound may be due to sub-optimal analysis, or sub-optimal choice of parameters in the algorithm"
> > >
> > > Even if the analysis is not tight in every aspect (which is always a possibility in theoretical work), this does not undermine the well-known fact: **non-vanishing neighborhood terms are unavoidable for any $(\varepsilon, \delta)$-DP algorithm.** This is not merely a consequence of our proof technique -- it is formally supported by the lower bound from Allouah et al. (2023) mentioned above.
> > >
> > > ---
> > >
> > > In summary, our work makes a novel and rigorous contribution by bridging three previously incompatible constraints -- DP, high-probability convergence, and heavy-tailed noise -- with fixed clipping. **We believe our results are both technically solid and practically relevant, and we kindly ask the reviewer to reconsider their assessment in light of these clarifications.**

---

> > > > ### Comment · Reviewer_hmsX · 2025-08-07
> > > > **followup on comparison with lower bounds**
> > > >
> > > > The lower bound in Allouah et al. (2023) decreases with the sample size at a $1/m$ rate (which is typical in ML theory). The upper bounds in this paper do not decrease with sample size, rendering them vacuous.

---

> > > > > ### Author Response · Authors · 2025-08-07
> > > > >
> > > > > Thank you for your follow-up. We would like to respectfully clarify the following.
> > > > >
> > > > > **On the lower bound.** While the lower bound in Allouah et al. (2023) decreases with $m$, it is **standard in ML theory and practice to treat $m$ as fixed** (reflecting a finite dataset size per client). Under this widely adopted regime, the lower bound becomes **independent of $K$** and thus non-vanishing in the number of iterations -- precisely mirroring the structure of our upper bounds. This regime aligns with typical assumptions in the federated learning literature.
> > > > >
> > > > > **On the finite-sum setting.** Furthermore, our analysis **does cover the finite-sum setting**, as we noted in lines 263–264 of the paper. As mentioned in lines 260–261, for unit mini-batch size, it is sufficient to set $\sigma_\omega \sim \frac{\lambda}{\varepsilon m}\sqrt{K \ln \frac{1}{\delta}}$. (In the original submission, we used $n$ to denote the number of samples; here we follow the notation from Allouah et al. (2023) and use $m$ for clarity.)
> > > > >
> > > > > Neglecting logarithmic factors, this implies that one can replace $\varepsilon$ with $\varepsilon m$ throughout Corollary 4.3. **As a result, the neighborhood terms in our bound decay with $m$, becoming vanishing in the limit $m \to \infty$**. Specifically, the term in question -- $\frac{LR^2 d}{\varepsilon^2}$ from equation (18) -- transforms into $\frac{LR^2 d}{\varepsilon^2 m^2}$, exhibiting the **same dependence on $d,\varepsilon$, and $m$** as the lower bound from Allouah et al. (2023). We will include this clarification and the corresponding corollary in the final version of the paper.
> > > > >
> > > > > **On the generality of the result and comparison to SOTA bounds.** Moreover, our bounds are derived under a **more general setting** -- with heavy-tailed noise, fixed arbitrary clipping levels, and high-probability guarantees, which to our knowledge have not been jointly addressed in prior work. In particular, our upper bounds match those of Koloskova et al. (2023) in terms of dependence on $d, \varepsilon, \delta$, and clipping level, confirming that the non-vanishing behavior is **not a weakness of our analysis**, but a reflection of fundamental DP limitations.
> > > > >
> > > > > ---
> > > > >
> > > > > We would be very grateful to hear your thoughts on these aspects of our response, especially the comparison with Koloskova et al. (2023) and the broader generality of our results.

---

> ### Comment · Reviewer_hmsX · 2025-08-07
>
> Thank the authors for clarifying the finite-sum results. The result is non-vacuous under such a setting, with $\sigma_w \asymp \frac{\lambda}{n \varepsilon} \sqrt{K}$. In ML problems, we always want the error to decay with more data, and this corollary seems the only non-vacuous result in the paper.
>
> Currently, this result is hidden in the text remarks, and is not rigorously proved (or even stated). I would suggest the authors to re-write the entire paper under this finite-sum setting and highlight the convergence rate in terms of $n$ (and choice of algorithmic parameters including $K$, as a function of $n$).
>
> Besides, it is well-known in DP optimization literature that private SGD algorithms can achieve small population-level loss for minimizing expected loss under random data (see e.g. Bassily et al., (2019), Private Stochastic Convex Optimization with Optimal Rates). So even if we use private SGD to minimize empirical loss, it is important to see how the population loss of the private clipped SGD output decay as a function of sample size $n$.

---

> > ### Author Response · Authors · 2025-08-07
> >
> > Thank you very much for your follow-up and thoughtful comments.
> >
> > We would like to clarify that while the result for the finite-sum setting is not explicitly stated in the current version of the paper, **we clearly described how it follows** from our general analysis (see lines 260–264 and our previous message), and we are fully committed to including this result, along with a corresponding corollary and formal statement, in the final version. This extension is **straightforward** and only requires a **minor adjustment** in the DP-noise scaling assumptions -- specifically, replacing $\sigma_\omega = \Theta\left(\frac{\lambda}{\varepsilon }\sqrt{K \ln\left( \frac{K}{\delta}\right) \ln\left( \frac{1}{\delta} \right)}\right)$ with $\sigma_\omega = \Theta\left(\frac{\lambda}{\varepsilon m}\sqrt{K \ln \frac{1}{\delta}}\right)$ and following the derivation in Corollary 4.3.
> >
> > Importantly, this result **directly follows from our existing proofs**. Our analysis is based solely on Assumption 2.4 regarding the stochastic gradient oracle, and that assumption already encompasses both the streaming and finite-sum settings. Therefore, the derivation for the finite-sum case is a **verbatim adaptation** of our current analysis.
> >
> > We thus **respectfully but firmly disagree** that the paper requires major rewriting. Instead, we view the final inclusion of the finite-sum corollary as a **natural and minor extension** of the current submission.
> >
> > Regarding the comparison with Bassily et al. (2019), we note that their work addresses a **completely different problem setup** -- specifically, population-level risk minimization with i.i.d. data and empirical risk formulation. In contrast, our setting is focused on **stochastic optimization using a streaming oracle**, with an emphasis on **high-probability convergence under heavy-tailed noise** and **arbitrary fixed clipping**. Due to these differences, a direct or fair comparison is not meaningful. We hope the reviewer appreciates that the **technical nature and goals of our results are distinct**.
> >
> > Lastly, we kindly ask the reviewer to also share their thoughts on the following two points:
> >
> > 1. Our **detailed comparison with Koloskova et al. (2023)**, whose bounds match ours in terms of $(d, \varepsilon, \delta)$ dependence.
> >
> > 2. The **generality of our setting**, which allows for arbitrary fixed clipping, heavy-tailed stochastic gradients, and high-probability guarantees -- **a combination not previously addressed in prior work** to the best of our knowledge.
> >
> > We sincerely thank the reviewer for their time and engagement.

---

> > > ### Author Response · Authors · 2025-08-07
> > > **Additional clarification about the DP noise**
> > >
> > > We would also like to add one clarification regarding the behavior of the DP noise parameter $\sigma_\omega$ in the finite-sum setting.
> > >
> > > As the number of samples per client $m \to \infty$, the DP noise level $\sigma_\omega = \Theta\left(\frac{\lambda}{\varepsilon m}\sqrt{K \ln \frac{1}{\delta}}\right)$ **tends to zero**, meaning that **no Gaussian noise is added in the limit**. This aligns with the privacy intuition: in the regime of infinite data, the influence of any individual sample vanishes, and thus **no explicit noise is needed to ensure differential privacy**. The algorithm remains differentially private by design, without requiring additional masking.
> > >
> > > Furthermore, we emphasize that **even in the case** $\sigma_\omega = 0$, our results remain **non-trivial and novel**. To the best of our knowledge, our work provides the **first high-probability convergence guarantees for Clipped-SGD under heavy-tailed noise with arbitrary fixed clipping levels**. This alone constitutes a meaningful theoretical contribution, independent of the DP setting.
> > >
> > > We hope this clarification further demonstrates the strength and novelty of our results.

---

### Official Review · Reviewer_Sg1D · 2025-07-02

**Clarity:** 3
**Significance:** 3
**Originality:** 3
**Rating:** 5
**Confidence:** 3

**Summary:**

This paper studies the high probability convergence rates of clipped SGD algorithm for smooth and bounded losses with bounded $\alpha \in (1,2]$ central moment (for some $\alpha$). Authors present a general convergence result for a clipped SGD, and extend that result for DP-SGD with clipped gradients. Authors obtain a convergence rate that improves the state of the art, even with a less restricting bounded moment assumption.

**Questions:**

1. Regarding the convergence rate in (9). You suggest that having a fixed clipping bound $\lambda$ leads to $\mathcal{O}(\sqrt{\ln{K / \beta} / K})$ rate. Can you clarify this? Would seem to me, that if with a particular $\lambda$ the learning rate is the third argument of eq. (8), you would cancel all the $K$ from eq. (9).
2. I'm a bit puzzled by the convergence rate arising from Corollary 4.3. The term (18) in the maximum seems to diverge w.r.t $K$, so I cannot see how you could obtain the rates presented in Table 1.
3. On line 262, you write $\epsilon = \mathcal{O}(q^2 K)$, but I think it should be $\epsilon = \mathcal{O}(q \sqrt{K})$.

**Ethical Concerns:**

["NO or VERY MINOR ethics concerns only"]

**Final Justification:**

Authors rebuttal addressed my concerns, which were mainly about the clarity issues in the paper. Hence I have increased my score from 4 to 5. I think the paper presents novel theoretical results that are interesting to the DP community.

**Limitations:**

yes

**Quality:**

3

**Strengths And Weaknesses:**

## Strengths
- Establishing convergence rates (especially high probability bounds) for DP-SGD is always a very interesting topic. It allows theoreticians to better understand the trade-offs of the algorithm, and helps practitioners to calibrate their privacy requirements to utility. (**Significance**)
- The obtained rates seem to improve the state of the art bounds, and authors have managed to do that by only assuming that the central moments for the gradients are bounded for some $\alpha \in (1,2]$ while the previous work assumed the same but $\alpha > 2$. This would make the presented bound more general. (**Significance**)
- Paper is in general very well written, and authors contribution is easy to understand. (**Clarity**)

## Weaknesses
- I have somewhat hard time recovering the main claim, the $\mathcal{O}(1/\sqrt{K})$ convergence rate, from the various theorems. Maybe it can be derived somewhat easily from the results, but it would help readability a lot if authors could connect the theorems to this claim. (**Clarity**)

---

> ### Author Rebuttal · Authors · 2025-07-28
>
> We thank the reviewer for their thoughtful comments, positive evaluation of our work, and recognition of its broader significance and clarity. Below, we address all the raised questions and concerns.
>
> > **Q1**: I have somewhat hard time recovering the main claim, the $\mathcal{O}(1/\sqrt{K})$ convergence rate, from the various theorems. Maybe it can be derived somewhat easily from the results, but it would help readability a lot if authors could connect the theorems to this claim.
>
> **A1**: We thank the reviewer for pointing this out. The correct convergence rate (to a neighborhood) is $\mathcal{O}(\sqrt{\ln(K/\beta)/K})$, which includes an additional logarithmic factor due to the high-probability nature of the bound. We will revise the statement in the Conclusion section accordingly, replacing $\mathcal{O}(1/\sqrt{K})$ with the more precise expression $\mathcal{O}(\sqrt{\ln(K/\beta)/K})$.
>
> > **Q2**: Regarding the convergence rate in (9). You suggest that having a fixed clipping bound $\lambda$ leads to $\mathcal{O}(\sqrt{\ln(K/\beta)/K})$ rate. Can you clarify this? Would seem to me, that if with a particular $\lambda$ the learning rate is the third argument of eq. (8), you would cancel all the $K$ from eq. (9).
>
> **A2**: Thank you for this insightful observation. As noted in lines 236–237 of the paper, we state that the convergence rate is $\mathcal{O}(\sqrt{\ln(K/\beta)/K})$, but we also clarify in lines 238–239 that this refers to convergence to a neighborhood whose size depends on $\lambda$, $\sigma$, $R$, and $\sigma_\omega$.
>
> The reviewer is correct that when the third term in the step-size bound (8) is the dominant one, then all the $K$ factors in (9) cancel out, and the optimization error does not decrease with $K$. In this regime, the algorithm does not converge to the global optimum but rather to a fixed neighborhood around it.
>
> This neighborhood is quantified explicitly in Corollary 4.2 (eq. (13)) and depends on the problem parameters, particularly on the fixed clipping threshold $\lambda$ and the noise level $\sigma$. Furthermore, when $\sigma_\omega = \mathcal{O}\left(\frac{\lambda}{\epsilon} \sqrt{K \ln(K/\delta) \ln(1/\delta)}\right)$, the neighborhood size also involves a growing term, as described in eq. (14).
>
> We will revise the final version to make this point more explicit and clearly distinguish the rate to the neighborhood from the size of the neighborhood itself.
>
> > **Q3**: I'm a bit puzzled by the convergence rate arising from Corollary 4.3. The term (18) in the maximum seems to diverge w.r.t. $K$, so I cannot see how you could obtain the rates presented in Table 1.
>
> **A3**: Excellent observation—thank you for catching this. You are correct that the last term in eq. (18) includes logarithmic factors in $K$, $\beta$, and $\delta$, and thus does not decrease with $K$. In fact, it slowly grows, i.e., polylogarithmically.
> However, in Tables 1 and 2, we provide the results for the non-DP case, i.e., when $\sigma_\omega = 0$. In the presence of non-trivial DP noise, Corollary 4.3 provides explicit upper bounds for two representative regimes corresponding to “large” and “small” clipping levels. While these do not always yield decaying rates with $K$, they still characterize the convergence to a meaningful neighborhood that depends on privacy and problem parameters.
> We note that similar bounds can be derived for other clipping regimes as well, though they lead to more complex expressions and were omitted for clarity. In the final version, we will clarify in the text and table captions that the tabulated results refer to the $\sigma_\omega = 0$ case.
>
>
> > **Q4**: On line 262, you write $\epsilon = \mathcal{O}(q^2 K)$, but I think it should be $\epsilon = \mathcal{O}(q\sqrt{K})$.
>
> **A4**: Thank you for pointing this out. In line 262, we refer to Theorem 1 from Abadi et al. (2016), which gives a sufficient condition for applying the moments accountant: $\epsilon \leq c_1 q^2 K$ for some constant $c_1$. This condition ensures that the moments accountant approach provides a valid $(\epsilon, \delta)$-DP guarantee.
>
>
> ---
>
> References:
>
> Abadi et al. “Deep learning with differential privacy”. Proceedings of the 2016 ACM SIGSAC conference on computer and communications security

---

> > ### Comment · Reviewer_Sg1D · 2025-08-05
> >
> > Thank you for addressing my concerns! Assuming the proposed changes to the clarity issues, I will raise my score.

---

> > > ### Author Response · Authors · 2025-08-05
> > >
> > > Thank you very much for your kind follow-up and for considering an increase in your score. We will make sure to incorporate all the proposed clarity improvements in the final version, including the revised explanation of convergence rates, improved cross-referencing between theorems and conclusions, and clearer notation in the tables. We truly appreciate your constructive feedback and engagement throughout the review process.

---

### Note · Authors · 2025-08-11

We thank all reviewers for their thoughtful evaluations and constructive engagement during the discussion phase.

---

## 1. Reviewer consensus

- Reviewers **Sg1D** and **5QfS** confirmed that all their concerns were fully addressed; the requested changes are minor and related to improving the clarity.
- Reviewer **hmsX** focused mainly on the **finite-sum setting**.

---

## 2. Finite-sum setting clarification

- Our analysis **already covers** the finite-sum case: the results follow **verbatim** from our current proofs, requiring only Assumption 2.4 on the stochastic oracle.

- The change is **minor** - essentially a direct corollary - and we are committed to including the explicit statement and proof in the final version.

- Under the standard ML regime with **fixed** $m$ (samples per client), the bounds are non-vanishing in $K$ and match the known lower bounds from *Allouah et al. (2023)* in their dependence on $d, \varepsilon, m$.

- In the limit $m \to \infty$, the DP-noise parameter $\sigma_\omega \to 0$, so no Gaussian noise is added; the algorithm remains private, and even in this case, our results are novel, providing the first high-probability guarantees for Clipped-SGD with fixed clipping under heavy-tailed noise.

---

## 3. Comparison to prior work

- Our bounds match those of *Koloskova et al. (2023)* in the non-convex DP-Clipped-SGD setting.

- The observed non-vanishing behavior is **inherent to DP** and not due to sub-optimal analysis.

---

## 4. Conclusion
The only remaining concern requires a **minor addition**, not a major revision. The main concerns have been resolved, and our work makes a **technically solid, novel, and relevant** contribution by jointly addressing:

- Differential privacy,

- High-probability convergence,

- Heavy-tailed noise, and

- Fixed clipping levels - a combination not covered in prior literature.

We kindly ask the AC to take into account the **technical strength, novelty, and broad relevance** of our work in the final decision.


---

**References:**

Allouah et al. “On the privacy-robustness-utility trilemma in distributed learning”. ICML 2023

Koloskova et al. “Revisiting gradient clipping: Stochastic bias and tight convergence guarantees”. ICML 2023

---

### Decision · Program_Chairs · 2025-09-17

**Decision:**

Reject

**Comment:**

This paper studies (differentially private) clipped SGD under heavy-tailed noise. It establishes general convergence results that strengthen and extend prior work in both the private and non-private settings.

The reviews were mixed: two reviewers were positive, while one reviewer (hmsX) was negative. The subsequent discussion did not lead to consensus. Instead, it reinforced the positive reviewers' opinions while leaving the negative reviewer's stance unchanged.

As is appropriate in such borderline cases, I carefully examined all materials and arguments. I found the authors' response to Reviewer hmsX convincing. In particular, I agree that in the private regime it is natural to expect convergence only to a neighborhood, and that non-vanishing bounds are to be expected in the general setting considered by the paper (which covers both stochastic approximation and finite-sum cases). Moreover, it is straightforward to adapt the results to the finite-sum setting and obtain rates that vanish with the number of samples $m$. This follows directly from the observation that the sensitivity of the clipped gradient scales as $O(\lambda/m)$. Therefore, I largely disagree with Reviewer hmsX's claim that the results are vacuous or that adapting them to the finite-sample case requires new derivations.

For these reasons, I recommend acceptance of the paper. However, all three reviewers raised concerns about the clarity and presentation. The authors provided convincing clarifications in their rebuttal and promised to incorporate them into the final version. **It is essential that the authors follow through and implement these improvements.**

===

As recently advised by legal counsel, the NeurIPS Foundation is unable to provide services, including the publication of academic articles, involving the technology sector of the Russian Federation’s economy under a sanction order laid out in Executive Order (E.O.) 14024.

Based upon a manual review of institutions, one or more of the authors listed on this paper submission has ties to organizations listed in E.O. 14024. As a result this paper has been identified as falling under this requirement and therefore must not be accepted under E.O. 14024.

This decision may be revisited if all authors on this paper can provide proof that their institutions are not listed under E.O. 14024 to the NeurIPS PC and legal teams before October 2, 2025. Final decisions will be communicated soon after October 2nd. Appeals may be directed to pc2025@neurips.cc.